# Human, Animal and Plant Health Benefits of Glucosinolates and Strategies for Enhanced Bioactivity: A Systematic Review

**DOI:** 10.3390/molecules25163682

**Published:** 2020-08-12

**Authors:** Sylvia Maina, Gerald Misinzo, Gaymary Bakari, Ho-Youn Kim

**Affiliations:** 1Smart Farm Research Center, Korea Institute of Science and Technology (KIST), Gangneung, Gangwon 25451, Korea; wairimusylvia@kist.re.kr; 2College of Veterinary Medicine and Biomedical Sciences, Sokoine University of Agriculture, Morogoro 25523, Tanzania; gerald.misinzo@sacids.org (G.M.); gaymary.bakari@sua.ac.tz (G.B.); 3SACIDS Africa Centre of Excellence for Infectious Diseases, Sokoine University of Agriculture, Morogoro 25523, Tanzania

**Keywords:** glucosinolates, glucosinolate hydrolysis products, natural compounds, secondary metabolites, bioactivity, improvement, bioavailability

## Abstract

Glucosinolates (GSs) are common anionic plant secondary metabolites in the order Brassicales. Together with glucosinolate hydrolysis products (GSHPs), they have recently gained much attention due to their biological activities and mechanisms of action. We review herein the health benefits of GSs/GSHPs, approaches to improve the plant contents, their bioavailability and bioactivity. In this review, only literature published between 2010 and March 2020 was retrieved from various scientific databases. Findings indicate that these compounds (natural, pure, synthetic, and derivatives) play an important role in human/animal health (disease therapy and prevention), plant health (defense chemicals, biofumigants/biocides), and food industries (preservatives). Overall, much interest is focused on in vitro studies as anti-cancer and antimicrobial agents. GS/GSHP levels improvement in plants utilizes mostly biotic/abiotic stresses and short periods of phytohormone application. Their availability and bioactivity are directly proportional to their contents at the source, which is affected by methods of food preparation, processing, and extraction. This review concludes that, to a greater extent, there is a need to explore and improve GS-rich sources, which should be emphasized to obtain natural bioactive compounds/active ingredients that can be included among synthetic and commercial products for use in maintaining and promoting health. Furthermore, the development of advanced research on compounds pharmacokinetics, their molecular mode of action, genetics based on biosynthesis, their uses in promoting the health of living organisms is highlighted.

Academic Editor: Jesus Simal-Gandara

## 1. Introduction

Glucosinolates (GSs) are natural, sulfur-rich anionic secondary metabolites, widely distributed in plants of the order Brassicales [1], mainly in the angiosperms families like *Brassicaceae*. Together with glucosinolate hydrolysis products (GSHPs), they are collectively described as mustard oil glucosides [2]. Only about 137 GSs have been characterized so far in plants [3]. Their core structure is composed of a β-D-glucosyl residue linked by a sulfur atom to a *cis*-N-hydroxyminosulfate ester, and a variable R group derived from a modified amino acid chain (which is the precursor used to group GSs into distinct classes). In plants, GS/GSHP compounds determine the distinct aroma, pungent flavors, and taste of foods [1].

GSs normally exist as intact compounds localized in vacuoles of different cell types. They are degraded to GSHPs by an endogenous glycosylated thioglucosidases enzyme known as myrosinase, which is physically separated in vacuoles of myrosin cells [4]. Myrosinase is activated upon cell disruption (e.g., during plant injury, feeding by herbivore/insects), metabolism by gut bacteria [5], usually in the presence of water. The enzyme hydrolyzes GSs thioglucoside bond producing glucose, sulfate and unstable aglycone moieties which are spontaneously rearranged to either isothiocyanates (ITCs), thiocyanates, epithionitriles, nitriles and oxazolidines, among others [1,3]. The type of GSHP compounds formed depends on the nature of the GS, reaction conditions (e.g., pH, presence of ions) and other compounds such as ascorbic acid and epithiospecifier proteins [6].

As natural chemicals that facilitate defense responses against different types of stresses (biotic and abiotic) in plant [7], GSs/GSHPs, have other diverse functions that have caused them to quickly gain in popularity as a subject of growing scientific interest. Plants utilize the GSs-myrosinase system, the “mustard oil bomb” [8] as a self-defense system against microbes and herbivores [9], and lowering of myrosin cell activity makes plants more susceptible to predators and insects like aphids [4]. In addition, compounds released from the hydrolysis of intact GSs by myrosinase enzymes are used as biocides/biofumigants in agriculture. Pharmacological studies have also shown that GSs/GSHPs have supplemental health promoting/beneficial properties as anti-inflammatory, antimicrobial, antioxidant, cholinesterase inhibitors and as cancer preventive agents in humans, while in the food industry these compounds are used in food preservation owing to their microbial inhibitory ability [10].

The majority of the biological activities of GSs are linked to their GSHPs [11], however, intact GSs also have the capability of modulating and impacting some biological systems [12]. These activities may be enhanced by the availability of more than one compound which display synergetic mode of action between or with other compounds [13]. Certain other factors which affect the bioactivity of these compounds include the plants’ GSs profiles (concentration and composition) [14] and hydrolysis enzyme actions [15]. Variations in plants’ GSs profiles and enzyme activity may occur among genotypes [16], cultivars, organs [17], accessions [18], varieties [19,20], growth stage and depending on environmental or growth conditions like photoperiods [21], temperature, water, nutrient availability.

Besides the naturally available sources of GSs, recently, the prediction of new, efficient bioactive GSs/GSHPs and their derivatives have been evaluated. Among them include ITC derivatives with higher antimicrobial activity [22], chemopreventive properties [23], herbicidal activity [24], antimicrobial activity [25], anti-tumor activity [26], ITCs with moderate anti-inflammatory [27,28,29] and ITCs with similar anti-proliferative activity as natural ITCs [30].

Furthermore, improvement strategies of natural GSs/GSHPs compounds using elicitors or biotic and abiotic stresses; which influence concentration, biological activity, GSs-myrosinase enzyme action, have also been assessed. In response to such factors, it has been shown that the expression profiles of various transcription factors related to GSs biosynthesis differ in various genotypes, which can be used to provide genetic diversity as well as phenotypic diversity in the GSs content [31].

Through the exploration of several studies, this review aims at providing up to date information on health benefits of different types of GSs/GSHPs, and the factors that affect their natural occurrence and bioavailability, for the various beneficial health roles they possess; so as to help in investigating the therapeutic values of these compounds in future.

## 2. Results

The search identified a total of 2198 articles that were retrieved from various databases, out of which 467 were considered potentially relevant based on their titles and abstracts in the initial screening. In a second screening, 282 articles were excluded after a thorough search through the text. Finally, 185 articles met the inclusion criteria, as shown in the PRISMA flow diagram (Figure 1).

A majority of these GSs identified from natural sources, were in plants of the *Brassicaceae*/*Cruciferae* family. Specifically, a large number belonged to the aliphatic group, followed by the aromatic/benzyl and then the indole group. The analysis of GSs/GSHPs was performed using high performance liquid chromatography (HPLC) analysis of desulfo-GSs derivatives according to the ISO-9167-1 method [32]. Among other GSs compounds mentioned, we identified that some only occur in specific families like glucocapparin in *Capparaceae* and glucomoringin (GMG) (4-α-L-rhamnopyranosyloxy) benzyl GSs in *Moringaceae* plant species.

Herbal preparations of GSs-rich extracts mostly focused on seeds and fully mature stages (flowers, leaves and roots) of the plants however some studies considered the utility of sprouts such as cabbage, broccoli, cauliflower, kale, brussels, radish and leaf mustard, probably due to their increased use as side dishes and salads in many countries.

The seeds, sprouts and young plant extracts, predominantly contained aliphatic groups of GSs (such as broccoli (*Brassica oleracea*) [14,33,34]), whose contents decreased with plant maturity whereas indole GSs contents increased. Furthermore, most of the plant, roots had higher concentration of total GSs than shoots.

Seed extracts that portrayed such benefits as chemopreventive, cytotoxic and antimicrobial include: - seed extracts of *Brassica juncea* L. containing 3- butenyl ITCs [35], *Lobularia libyca* [36], *Carica papaya* L. rich in benzyl GS [37], mustard seed powder [38], *Sinapsis alba* L. and *Sinapsis nigra* L. with high content of sinalbin and sinigrin, respectively [39]. GMG isolated from *Moringa oleifera* L. seeds possessed chemopreventive [40,41,42], insect larvae deterrence [43], antibacterial [44] and anti-inflammatory activity [45]. Volatiles in *Lunaria annua* L. seed extracts displayed cytotoxicity activity against lung cancer cells [46].

Seed meal of *Brassica juncea* suppressed *Meloidogyne incognita* nematodes individually and in specific combination with *Sinapsis alba* [47] while *Camelina sativa* L. seed meal exhibited anticancer activity [13]. Elsewehere, meadowfoam (*Limnanthes alba*) seed meal and extracts had photoprotective properties [48] biopesticidal activity against soil borne pathogens [49] and phytotoxic activity [50].

Volatiles in leaves extracts of *Degenia velebitica* [51], in aerial parts extracts of *Cardaria draba* L. [52] and in *Aurinia leucadea* [53] possessed antimicrobial activity and leaves extracts of *Lepidium latifolium* L. [54] had cytotoxic activity. The leaves extract from *Brassica* species displayed nematicidal activity [55] and chemopreventive ability [11,56,57]. The *Wasabi japonica* leaf extracts showed anti-neuroinflammatory effects [58], while those in horseradish (*Armoracia rusticana*) had antimicrobial, spasmolytic, cytotoxic activities [59,60]. Essential oils extracted from caper leaves and flower buds had anticarcinogenic activity [61]. Tumor growth inhibition and cytotocicity effects on human cancer cells were also seen in leaf extracts of *Brassica carinata* [62] and *Eruca sativa* [63] and GMG from leaves extracts of *M. oleifera* exhibited antioxidant activity [64].

Root extracts of horseradish exhibited nematicidal activity against *Meloidogyne incognita* [65], while those from turnip (*Brassica rapa* L.) had a stimulatory effect in bone formation in rats [66], while from *Bunias erucago* showed chemopreventive activity [67]

Sprouts extracts of *Raphanus sativus* L. induced apoptosis and cytotoxicity in human hepatocarcinoma cells [68] and human breast cancer cells [69]. The sprout extracts of Chinese kale (*B. oleracea* var. *alboglabra*) exhibited antioxidant activity [70]. Raphasatin in radish sprouts was also identified to be more potent as a chemopreventive compound than its corresponding GSHP [71]. Table 1 summarizes the health benefits of GSs including the herbal extracts mentioned above.

## 3. Benefits of Natural and Synthetic GSs and GSHP

### 3.1. Biofumigation/ Biocidal Activity and Pest Management

Natural GSs and their derivatives possess biocidal activity and show toxicity to a range of soil borne as well as plant pathogens and pests [55]. GSs rich plant sources and extracts with the ability to inhibit the growth of pathogens, offer the opportunity to explore them in controlling many plant diseases and as potential bio fumigants [72]. GSs and GSHPs volatiles, too, such as ITCs, have at low concentrations been used to control plant pathogens and/or are included as active ingredients among synthetic commercial nematicidal compounds [6]. These compounds possess bioherbicidal, larvicidal, soil fungicidal, insect feeding deterrence, nematicidal activities, and growth retardation [49,50,65,73,74,75,76,77,126,128]. Their activity and effectiveness mainly increase with exposure time and in a dose-dependent manner [73,126].

Disease suppression in soil has been done using rotations of GSs-rich green manure. *Brassica juncea* L. seed meal soil supplement displays greater nematotoxicity to *Meloidogyne incognita* pathogens and in combination with *Sinapsis alba* seed meal, the manure helps deter nematodes and suppresses weeds in pepper plants [47]. A similar combination displays potential in managing *Verticillium* wilt and soil-borne diseases [136]. *B. juncea* L. leaf green manure rich in 2-propenyl GSs also exhibit high nematicidal activity both In vitro and in soil microcosms [55]. *Boscia senegalensis* seed wastewater, rich in methyl ITCs, has allelopathic effects which cause inhibition of germination of weeds seeds [78] while non-volatile ITCs like moringin facilitates feeding deterrence of Brassicales specialists by exhibiting antifeedant activities [43].

Besides, synthetic ITCs also display potential fungicidal structure-activity relationship and they have prospective acceptability as alternatives to traditional fungicides. They include *p*-nitrophenyl which inhibits *Rhizoctonia solani* and *Erwinia carotovora* pathogens [25].

The bioprotective role of some GSs is successively promoted by the action of other compounds present in the plant. For instance, indole GSs together with the phytoalexin camalexin cause *Arabidopsis thaliana* resistance to the oomycete pathogen *Phytophthora brassicae.* While indole GSs individually inhibit host penetration and play initial defensive roles, the phytoalexin camalexin enhances the defense at later stages by adding to this initial protection [137].

### 3.2. Antimicrobial Activity

Microbial drug resistance, pathogen resistance, food spoilage, clinical and oral microbial pathogens are becoming a global problem. Thus, there have been increasing efforts to evaluate natural products as a source of compounds against these agents. GSs/GSHPs are among the most important natural products whose antimicrobial mode of action has been characterized [10,36,82,83,84].

The antimicrobial activity of plant extracts possessing therapeutic potential relies on the presence of volatiles, among which GSs and GSHP compounds have been well identified. Studies focusing on these compounds and their antimicrobial activities show that active compounds affect pathogens’ membrane integrity [127], intracellular potassium release [10], inhibit growth in pathogens [22,36,39,51,52,53,54,79,80,83,87,89,90], kill pathogens [81,82,88] and change physiochemical surface properties and charges. Inhibitory effects of GSs and GSHPs can be non-selective [59] or may vary between different pathogens, as illustrated by allyl and phenylpentyl ITCs in *A. rusticana* which has higher antifungal activity than antibacterial activity [80].

Natural and synthetic volatile ITCS compounds have also been shown to inhibit the growth of drug-susceptible and multidrug-resistant (MDR) pathogens at concentrations comparable to the commonly used antimicrobial drugs [22]. Furthermore, they show even higher activity against MDR strains compared to susceptible strains, as in the case of *Mycobacterium tuberculosis* [22].

The effectivity of GSHPs as antimicrobial compounds can be increased by combining them with less efficient antibiotics [84,127], (which results in synergism) and/or having a mixture of several ITCs that display a broad-spectrum antimicrobial activity even to MDR pathogens [83,85]. Bioactivation of GSs with myrosinase enzyme also maximizes their effectiveness and antimicrobial activity, as demonstrated by enzyme bioactivated GMG activity against *Pseudomonas aeruginosa* and *Staphylococcus aureus* [44]. Food-borne bacteria synthesize myrosinase enzyme, which also influences the conversion of GSs to their GSHPs and increases their potency as antimicrobials [86].

### 3.3. Antioxidant Activity

Reactive oxygen species and free radicals produced by several metabolic processes have very high potential to harm cells through oxidation. Their continued presence speeds up aging resulting in age-related illnesses [138]. Several antioxidants studies show that GSs/GSHPs can scavenge and eliminate the damaging effects of the reactive oxygen species directly or indirectly [64].

The various GSs abundant plants showing potential antioxidant activity include Chinese cabbage (*B. rapa* L.) varieties [19], *B. juncea* L. (kimchi) [139], *Isatis canescens* flower buds [122], leaves of *Eruca sativa* Mill [140], *Brassica olerecea* L. sprouts [141], *B. juncea* L. leaves and seeds extracts [91], and curly kale leaves juice [92]. Plant extracts with antioxidant activity can be optimized by choosing appropriate conditions for their processing and storage, as demonstrated in fermented kimchi [139]. Elsewhere, synthetically obtained phenyl ITCs and their derivatives also display good antioxidant activity [129].

### 3.4. Anti-Inflammatory Activity

GSs and GSHPs have shown great potential as anti-inflammatory agents through their ability to suppress inflammatory mediators independently or with other substances [131]. Phenethyl ITCs down-regulate the nuclear factor κB pathway, subsequently inhibiting transcription of genes involved in inflammatory effects, on a mouse ear topically applied with an inflammation-inducing substance [130]. In combination with 3-methoxyphenyl ITCs, phenyl ITCs cause approximately 99% inhibition on the human cyclooxygenase-2 enzyme [129].

The anti-inflammatory activity of GSs/GSHPs is presented in various ways for different cells. Plant extracts of *Wasabi koreana*, containing allyl ITCs, protect neuronal cells from activated microglia induced toxicity by inhibiting activation of nuclear factors [58] while sulforaphane isolated from broccoli sprouts improve cognitive function among schizophrenic patients [95]. Myrosinase- activated GMG [45] and glucoraphanin [97] act as potent anti-inflammatory agents by inducing inflammatory pathways.

Anti-inflammatory activity may be established in a concentration-dependent manner, as shown by sinigrin which suppress the production of inflammatory mediators in activated macrophages by inhibiting lipopolysaccharide-induced nitric oxide pro-inflammatory mediators [93]. To achieve maximal activity it is worth activating compounds as in the case of bioactive glucoraphanin that preserves tight junctions’ integrity preventing dysfunctional blood-brain barriers in autoimmune encephalomyelitis, causing multiple sclerosis [96]. Furthermore, synthetically produced GSHPs are worth evaluating for their activity like indole GSs [28], and aromatic GSs [27] which have an inhibitory effect in the secretion of tumor necrosis factor-α in human monocytic leukaemia T peripheral helper-1 cells stimulated with lipopolysaccharides.

### 3.5. Cholinesterase (ChE) Inhibitory Activities

GSHPs, mainly volatile ITCs, demonstrate potential in anti-hepatotoxic activity. They include gluconapin, glucoerucin, and glucoraphanin that display above 53% acetylcholinesterase (AChE) inhibitory activity in a dose-dependent manner [98]. Hydrodistillate extract from *Bunias erucago* flowers, rich in glucosinalbin, shows a best AChE inhibition of 40.9% while roots extract displays a butyrylcholinesterase inhibition activity of 54.3% [67].

In addition to natural GSs, synthetic GSHPs and their derivatives also show good cholinesterase inhibitory activity. This has been illustrated in phenyl ITCs and its derivatives that display the most promising inhibitory activity with potential applications in treating Alzheimer’s disease [129].

### 3.6. Cytotoxic and Anti-Cancer Activity

The mechanisms contributing to the anti-tumor and/or chemopreventive properties of GSs/ GSHPs are varied, as shown in various studies included in this review. Compounds may act by inducing/modulating systems of carcinogen detoxifying/metabolizing enzymes [103,117] which include increased detoxification, they may also act by upregulating phase II enzymes such as glutathione-S-transferase [104,123,125] and quinone reductase [11,13,18,71,106,107]. These mechanisms may be organ specific [123], may depend on the compound enantiomer involved [103], on the amounts of compounds [142], and on the availability of various compound which produce synergistic action [13].

GSHPs-ITCs may inhibit enzyme activity, as evidenced in 2-propenyl and 3-butenyl ITC that inhibit cytochrome P450 1A enzyme activity [99]. The compounds may also display selective anticancer activity that results in the inhibition of deoxyribonucleic acid replication in cancer cells as shown by sulforaphane and phenethyl ITCs, causing more double strand breaks in prostate cancer cells [110].

Furthermore, GS/GSHP compounds exhibit radical scavenging ability and cytotoxic effects in various cancer cells. Compounds like butenyl thiocyanates, allyl, phenethyl ITCs [100], and GMG GSs [133] reveal such effects. The compounds’ effectivity lies in their ability to inhibit cell proliferation, [37,39,61] arrest the cell cycle [38], causing morphological changes, cytotoxicity, and cell apoptosis [35,36,40,41,54,56,59,69,101,113,115,119]. These activities may be dose dependent [37] and may also be defined by the chemical nature of the compound [143]. Sometimes, the dynamism of these activities occurs through synergic effects [105] of various compounds present in an extract and the particular extract’s vigor is affected by the method of preparation [102].

Some GSs/GSHPs display their potential by possibly inhibiting compounds or activities related to cancers. Among them include, phenyl ITCs that inhibit transcriptional activity regulated by androgen receptor modulating growth of prostate cancer cells through down-regulation of an androgen receptor-regulated gene [134]. ITCs may influence changes and activity of compounds that are normally overexpressed during cancer such as histone deacetylase [112,114] or compounds which may cause intoxication such as doxorubicin [124].

GSs/GSHPs exhibit the ability to suppress healthy cell invasion and abolish the growth of tumors by affecting pathways that adjust carcinogen metabolizing compounds. These include sulforaphane ITCs, which down regulates the expression of matrix metalloproteins [108]. Such ITCs may activate the antioxidant response element electrophile response element/antioxidant responsive element, modulating estrogen signaling in different tissues that leads to induction of the inhibitory effects of estrogen in breast cancer cells [109], while in cortical neurons, they activate the extracellular signal regulated kinase pathway, transcriptional factors, and upregulate detoxification enzymes [111]. Benzyl and phenethyl ITCs show significant inhibition of the proteasome activity and also suppress the growth of multiple myeloma cells [135].

The photoprotective properties of 3-methoxybenzyl ITCs and 3-methoxyphenyl acetonitrile, in *Limnanthes alba,* display their anti-photocarcinogenic and anti-photoaging properties. These ITCs induce human skin cell proliferation activity against ultraviolet B rays and prevent hyperplasia induced by these rays in reconstructed skin epidermal tissue [48].

### 3.7. Other Activities

Sulforaphene and raphasatin ITCs reduce the risks of sugar-related diseases. In addition, these compounds cause spargel expression, which modulates energy metabolism, inhibiting α-amylase, and α-glucosidase in vitro [120]. Elsewhere, sulforaphane and phenethyl ITCs suppress urinary mutagenicity activity as illustrated on rats treated with oral doses of heterocyclic amine 2-amino-3-methylimidazo[4 ,5-f]quinoline, a food carcinogen. Long term intake of the ITCs decreases urinary mutagenic activity, implying enhanced/modulation of the metabolism of the quinoline [121].

Sinigrin, a common GS in most cruciferous vegetables, exhibits the great potential of inhibiting non-enzymatic aglycation [119], while its GSHP allyl ITCs has hepatoprotective roles (reduction of hepatic fibrosis). In rats exposed to carbon tetrachloride (CCl_4_), the ITCs reduce alanine aminotransferase and aspartate aminotransferase activity, activate macrophages and regulate Kupffer cells [118].

The potential stimulatory effect of GSs in bone formation, and as natural sources of promising neutraceutical agents has been shown in *B. rapa* L. roots GSs, that show stimulatory effects on human osteoblast-like MG-63 cell differentiation. Administration to young rats increases serum osteocalcin as well as some bone microstructural parameters [66]. In addition, 5-phenylpentyl ITCs isolated from *A. rusticana* exert significant spasmolytic activity on rats’ distal colon [59].

Functional foods based on GSs display parasiticidal potential against *Lepeophtheirus salmonis* lice, ectoparasites that infect Atlantic salmon. In an in vivo study that involved various feeding trials on fish, beneficial effects on expression of genes functioning in detoxification and regulation of iron metabolism in several tissues are shown. Such properties affect the availability of heme to lice, deterring them from infesting the fish [144].

### 3.8. Effects of GSs/GSHPs Used in Preclinical and Clinical Studies

Clinical and preclinical studies evaluating GSs/GSHPs components and diets provide the basis for further explaining the mechanism of action of these components. Broccoli sprouts, rich in sulforaphane administered in a pilot study on melanoma patients display chemoprevention ability [145] and shows favorable effects in reduction of serum insulin while at the same time enhances insulin resistance in type 2 diabetes patients, therefore improving their glycaemic control and decreasing risk factors of cardiovascular diseases [146]. Broccoli soup consumption also reduces the risk of prostate cancer progression [147] and daily administration of free sulforaphane shows promising evidence in managing biochemical recurrence after radical prostatectomy in patients with prostate cancer [148]. Similarly, in asthmatics, sulforaphane enhanced bronchoconstrictor effect, improving major defects and even mild asthma [149] and in schizophrenic patients, sulforaphane from broccoli sprouts helped in improving their cognitive function [95].

Clinical studies, however, require a good design that follows a standardized protocol which includes selecting a representative sample size and various doses [148] of the phytochemicals to minimize bias and allow generalization. Likewise, masking and the duration of treatments [146,150]/interventions/exposure/supplementation [151] should be well considered to confirm specific effects of the components. Variabilities in concentrations of GSs/GSHPs consumed need to be minimized in the diet within these studies. Furthermore, research on the mechanism of action, probably in the biomarker-based approach of GSs/GSHPs is highly recommended to understand various relationships between exposure and risks of the phytochemicals as reported in some studies [152,153]. Table 2 summarizes some preclinical and clinical studies evaluating the health benefits of GSs and GSHPs.

### 3.9. Improving Plant GSs, GSHPs Content, Their Bioavailability, and Bioactivity

There are several factors that affect GSs contents/composition as well as their biological activities. These include (a) post-harvest treatment of GSs rich plant [154,155,156], (b) preparative activities like boiling and nature of vegetables (like those with thick waxy tissues have minimal leaching of GSs into the boiling medium) [157]; (c) extraction solvent polarity [46,158]; and (d) extraction technique [39,60,132,159,160].

Salt stress, in a concentration-dependent manner, is a potential inducer of producing these health-promoting compounds. Sodium chloride treatment increases glucoraphasatin on radish sprouts [161], and sulforaphane in broccoli sprouts [162]. Salt stress in combination with elevated carbon dioxide increasing sulphur cellular partitioning and metabolism which affects the GSs-myrosinase system [163]. Similarly, sulphur salts supplementation during the cultivation of vegetables increases GSs content and their biological activities [68].

Exogenous application of phytohormones triggers/manipulates GSs, GSHPs, and their related gene expression profiles. Among the phytohormones used include methyl jasmonic acid (MeJA) [164], gibberellic acids (GA3) in combination with glucose [70] and jasmonic acid (JA) [165]. A 250 µM MeJA spray treatment on *Brassica napus* var. *pabularia* seedlings [166] and *B. oleracea* var. *green magic* before harvesting [164] induces GSs. As GSs content increase during MeJA treatment, there is a simultaneous enhancement of biological activity [57,167,168,169,170]. The GSs content changes observed in plant may be cultivar specific [171]. A combination of 5 µM GA_3_ and 3% glucose in Chinese kale sprouts increases GSs content and enhances their biological activities [70]. Also, JA treatments enhance the accumulation of aliphatic GSs, increase myrosinase activity and GSHP production [165].

Overexpression of genes involved in the breakdown of GSs defends plant against pathogens. In *A. thaliana,* overexpression of myrosinase gene TGG1, affects the metabolism of GSs compounds enhancing plant stomatal defense against *Pseudomoas syringae* bacteria [172].

The treatment of plants with trace/essential elements affects the GSs-myrosinase system. Selenium treatment for instance, increases sulforaphane in broccoli cultivars sprouts, enhancing their myrosinase activity [34] while in *R. sativus* an increment in glucoraphanin is observed [173]. In *A. thaliana* and *B. oleracea* Var. *italica*, ammonium nutrition stimulates the accumulation of GSs and induces myrosinase activity as well [174]. Managing plants using chemicals and light may increase GSs content, for instance 6-benzylaminopurine [175] alone and in combination with 1-methyl-cyclopropene [176] increases GSs in broccoli and green light emitting diode lights increase broccoli’s florets’ total GSs content and sulforaphane [177]. Elsewhere, GSs in plants may accumulate through natural occurrences, in response various stresses like yearly seasonal variations that cause increased temperatures, water stress [168], larval/herbivore infestation [166,178,179], ozone fumigation [180], and biotechnological transformations [172,181].

Dietary intake of GSs rich products is positively correlated to their health benefits and biological activity. Their bioavailability can be enhanced by increasing the hydrolysis enzyme activity and/or treating extracts in mild heat [182]. Continuous consumption of GSs rich foods is exogenously treated with myrosinase [183,184,185,186] or intake of diverse GSs rich foods which increase the activity of the GSs degradation enzyme in the gut results into the bioavailability of GSs/GSHPs [187]. The content of these compounds in prepared and processed food is essential and therefore, post-harvest processing techniques, food processing, and treatment methods that affect the contents of GSs and the biological activities should be carefully chosen. Pasteurization [92], high pressure [188,189] treatments and freezing [190,191], thermal boiling [133,157,192], steaming [193], microwaving [46,194,195], storage time [139], storage temperatures [196] affect to GSs contents. On the other hand, blanching is considered to prevent loss of phytochemicals [102] in vegetables. Table 3 and Table 4 summarize the factors and treatments which affect the availability and activity of GSs/GSHPs in plant or diet and the hydrolysis activity of myrosinase enzyme. Several substances that have been used in the elicitation of GSs/GSHPs have shown not only to induce physiological changes but also stimulate various biological activities, as illustrated.

## 4. Materials and Methods

### Search Strategy and Selection Criteria

Peer-reviewed articles were searched using keywords in various scientific databases, including PubMed, ResearchGate.net, Wiley Online Library, and Google Scholar, between March 2010 and March 2020. The search terms used to obtain articles for this review were “glucosinolate” OR “glucosinolate extract” OR “plant glucosinolate” OR “glucosinolate isolate” OR “pure glucosinolate” OR “synthetic glucosinolate” OR “glucosinolate hydrolysis product” OR “Glucosinolate degradation product” OR “mustard oil glucoside” OR “myrosinase end product” OR “isothiocyanate” OR “benzyl glucosinolate” OR “indole glucosinolate ” combined with “profile” OR “availability” OR “variability” AND “activity” OR “Bioactivity” OR “health benefits” OR “specific biological activities” AND “improvement” OR “elicitor”. Data extraction included the specific glucosinolate/ glucosinolate hydrolysis products, their sources, biological activities, mode of action, availability, strategies used to increase their content availability and activity. Articles published in a non-English language, before and/or after the set period, retracted from the databases, having self-reported outcomes, and reviews and editorials, were excluded. The review focused on the peer-reviewed original studies with a particular outcome of interest, depending on the study objective.

## 5. Conclusions

GSs/GSHPs offer a wide variety of health benefits, including disease prophylactic and therapeutic effects. This review shows the usefulness of these compounds in preventing and reducing disease progression in humans and animals, their biocidal, biofumigation capabilities in plants, and their antimicrobial use in the food industry as food preservatives.

Although their composition and concentration vary in various crop species, organs, cultivars, and at different stages of development, sometimes in response to both abiotic and biotic factors, strategies to improve specific compounds have been successful, leading to improved crop varieties with both nutritional and pharmacological benefits. Biological activity assays on GSs-rich extracts exhibit a positive correlation between concentrations of GSs/GSHPs-related benefits in organisms. Taking this into consideration, the bioavailability of these compounds should be well maintained in their sources through choosing and having handling, preparative techniques, extraction methods that maintain them.

This review also reveals the necessity to maintain GSs content as bioactive compounds, to explore and improve GSs rich plant as a source of these natural compounds, which have potential as active ingredients among synthetic and commercial products to maintain and promote health. Considering that most research evaluating these natural compounds mainly focuses on the plant of *Brassicaceae* family, there is furthermore a need to probe the compounds even more in vivo studies, to understand their primary mechanism of actions and their molecular targets should also be emphasized.

## Figures and Tables

**Figure 1 molecules-25-03682-f001:**
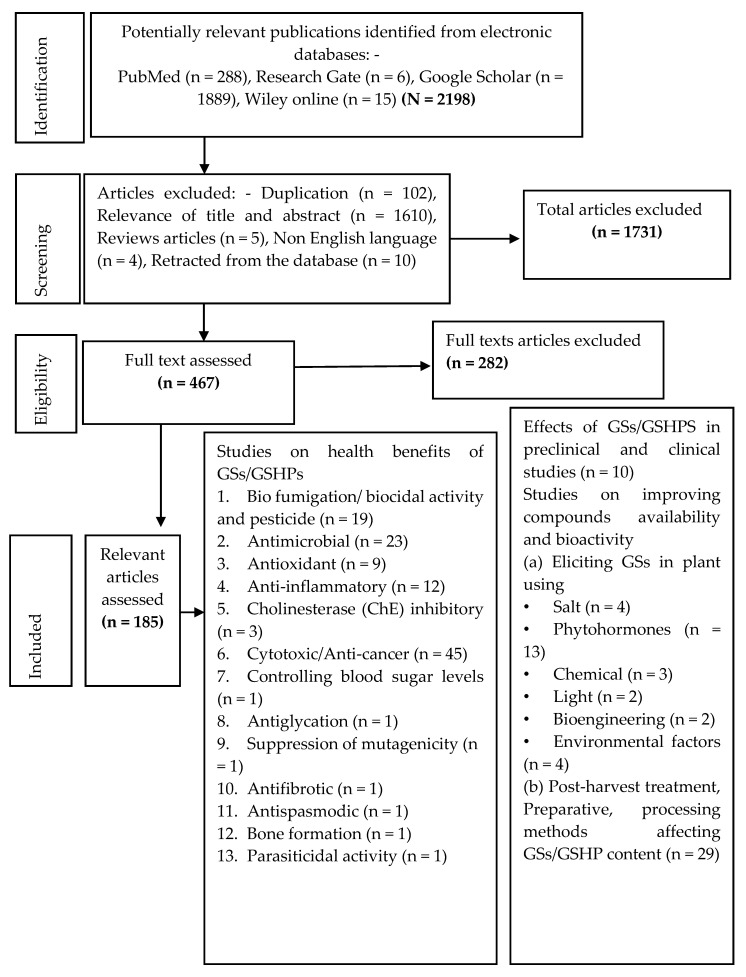
This figure shows the flowchart used in the article selection process.

**Table 1 molecules-25-03682-t001:** Various health benefits of GSs and GSHPs.

GSs	Hydrolysis Product	Biological Activity	Source	Organism/Pathogen/ Cell	Reference
**Aliphatic GSs**					
**Biocidal/ Biofumigation Activity**					
Glucoraphanin, glucoiberin glucobrassicanapin, glucoerucin	Sulforaphane 4-pentenyl ITCs	Biocide	Synthetic		[72]
Sinigrin	Allyl ITCs/2-propenyl ITCs	Herbicidal Larval feeding deterrent	Synthetic L. *latifolium* L.	*Cyperus esculentus* weed *Plutella xylostella* pest	[73,74]
		Nematicidal activity	*A. rusticana*	*M. incognita*	[65]
		Insecticidal	synthetic	*Helicoverpa armigera*	[75]
		Biocide	Synthetic		[72]
		Nematicidal activity	*B. juncea* L.	*Globodera pallida*	[55]
		Biofumigation	*A. thaliana*	*Verticillium longisporum*	[76]
		Fungitoxic activity	Synthetic	* Sclerotinia sclerotiorum *	[77]
Gluconapin	Butyl ITCs	Fungitoxic activity	Synthetic	* S. sclerotiorum *	[77]
		Biocide			[72]
Glucocapparin	Methyl ITCs	Fungitoxic activity	Synthetic	* S. sclerotiorum *	[77]
		Weed germination inhibition	*B. senegalensis*		[78]
**Antimicrobial activity**					
Sinigrin		Antimicrobial	*S. alba* L. and *S. nigra* L.	*S. aureus, Streptococcus pyogenes, Bacillus cereus, E*scherichia *coli, P. aeruginosa, Candida albicans*	[39]
		Antibacterial	*B. oleracea*	*Bacillus cereus, E. coli, Salmonella typhimurium,* Methicilin resistant *S. aureus*	[79]
		Antifungal	*A. rusticana*	*Trichophyton rubrum, Trichophyton mentagrophytes, Microsporum canis, Epidermophyton floccosum*	[80]
		Antimicrobial	*A. rusticana*	*C. albicans, Fusobacterium nucleatum*	[81]
		Antimicrobial	Cruciferous plant	methicillin resistant S*. aureus*	[82]
		Antimicrobial and cytotoxic	*L. latifolum* L.	*Listeria monocytogenes, Acinetobacter baumannii, C. albicans, S. aureus, S. Typhimurium, E. coli, P. aeruginosa*	[54]
			Synthetic	*Haemophilus influenzae, Moraxella catarrhalis, Serratia marcescens, Proteus vulgaris*, and Candida species *E. coli, P. aeruginosa, S. aureus* and *L. monocytogenes* *P. aeruginosa*	[10,83,84,85]
		Antimicrobial	Synthetic	*E. coli*	[86]
		Antimicrobial activity	*B. oleracea*	*E. coli, Klebsiella pneumoniae*	[87]
		Antimicrobial	*A. rusticana*	*C. albicans, F. nucleatum*	[81]
		Bactericidal	synthetic	*Campylobacter jejuni*	[88]
Glucoraphanin	4-(Methylsulfanyl)butyl ITCs & 5-(methylsulfanyl)pentanenitrile	Antimicrobial	*C. draba* L.	*E. coli, K. pneumonia,* *Enterobacter sakazakii, P. aeruginosa, Cronobacter spp., S. aureus, Rhizopus stolonifer*	[52]
		Antibacterial	Synthetic	Pig aeromonas intestinal bacteria	[89]
Gluconapin	3-Butenyl ITCs	Antimicrobial	*A. leucadea*	*Bacillus cereus, C. albicans, Penicillium sp., R. stolonifera, P. aeruginosa*	[53]
		Antimicrobial activity	*B. oleracea*	*E. coli, K. pneumoniae*	[87]
		Antimicrobial	*A. rusticana*	*C. albicans, F. nucleatum*	[81]
Glucoerucin	4-(Methylsulfanyl)butyl ITCs, 5-(methylsulfanyl)pentanenitrile	Antimicrobial	*C. draba* L.	*E. coli, K. pneumonia,* *E. sakazakii, P. aeruginosa, Cronobacter spp., S. aureus, R. stolonifer*	[52]
		Antimicrobial activity	*B. oleracea*	*E. coli, K. pneumoniae*	[87]
		Antimicrobial	*L. libyca*	*C. albicans* and *P. aeruoginosa*	[36]
Glucobrassicanapin		Antimicrobial	*Aurinia sinuate*	Gram positive, negative bacteria and fungi	[90]
			*A. leucadea*	*B. cereus, C. albicans, Penicillium sp., R. stolonifera, P. aeruginosa*	[53]
Glucoalyssin		Antimicrobial	*A. leucadea*	*B. cereus, C. albicans, Penicillium sp., R. stolonifera, P. aeruginosa*	[53]
Glucoiberverin		Antimicrobial activity	*B. oleracea*	*E. coli, K. pneumoniae*	[87]
Glucoberteroin	6-(Methylsulfanyl) hexanenitrile	Antimicrobial	*D. velebitica*		[51]
	aliphatic ITCs and their derivatives	Antimicrobial	Synthetic	*Mycobacterium tuberculosis*	[22]
**Antioxidant activity**					
Sinigrin	Allyl ITCs	Antioxidant	*B. juncea* L.		[91]
			*B. rapa* L.		[19]
			Curly kale leaves		[92]
Gluconapin		Antioxidant	*B. juncea* L. *B. rapa* L.		[19,91]
Glucoalyssin, progoitrin, glucobrassicanapin,		Antioxidant	*B. rapa* L.		[19]
Glucoiberin		Antioxidant	Curly kale leaves		[92]
**Anti-inflammatory activity**					
Sinigrin	Allyl ITCs	Anti-inflammatory	Synthetic		[93]
			*W. koreana*		[58]
Glucoraphanin	Sulforaphane	Inflammatory Prophylactic	Broccoli sprouts		[94]
		Anti-inflammatory	Broccoli sprouts		[95]
			Tuscan blackkale		[96]
			*B. oleracea*		[97]
Neoglucobrassicin		Anti-inflammatory	synthetic		[28]
**Cholinesterase Inhibitory Activity**					
Gluconapin, Glucoerucin Glucoraphanin	But-3-enyl ITCs erucin sulforaphane	Acetylcholinesterase inhibitory activity	*Alyssoides utriculata*		[98]
**Cytotoxic Activity and chemoprevention**					
Sinigrin	Allyl ITCs	Quinone reductase activity	*A. rusticana*	Hepa1c1c7 murine hepatoma cells	[18]
		Cytotoxicity	Synthetic	MCL-5 cells	[99]
		Anti-cancer, radical scavenging ability and increase ROS	*B. juncea* var. *raya*	human breast cancer (MCF-7, MDA-MB-231), prostate cancer (PC-3), lung cancer (A-549), cervical cancer (HeLa) and colon cancer (HCT 116) cells	[100]
		Anti-cancer	Mustard seed powder	bladder cancer cell lines and orthotopic rats model bladder	[38]
		Cytotoxic	*L. latifolum* L.	Glioblastoma LN229 cells	[54]
		Anti-cancer, anti-melanoma	*E. sativa* seed oil	HepG2 human liver carcinoma and BB16F10 mice melanoma cells	[101]
		Anti-tumor	Collard	Human MCF-7, HeLa cells	[102]
		Anti-proliferative	*S. alba* L. and *S. nigra* L.	colon HCT 116 and HT-29 cells	[39]
		Anti-tumor	Collard	Human MCF-7 and HeLa cells	[102]
Glucoraphanin	Sulforaphane	Chemoprevention	*B. oleracea*	Liver and lung cells	[103]
		Chemoprevention	Japanese Daikon	Rats liver and lung	[104]
		Anti-multiple myeloma activity	Synthetic	Myeloma cells	[105]
		Chemoprevention	*B. oleracea*	Human HepG2 hepatoma cells	[11]
		Anti-cancer, anti-melanoma	*E. sativa* seed oil	HepG2 human liver carcinoma, BB16F10 mice melanoma cells	[101]
		Quinone reductase activity	Chinese kales	Hepa 1c1c7 murine hepatoma cells	[106]
		Quinone reductase activity	*E. sativa* Mill	murine hepatoma Hepa1c1c7 cells	[107]
		Anti-metastatic	Synthetic	MDA-MB-231(breast), Caski (cervical), A549 (lung), and U2OS (osteosarcoma) cell lines	[108]
		Anti-cancer	Synthetic	breast cancer cell	[109]
		inhibit cancer cells DNA replication	Synthetic	PC-3 prostate cancer cells	[110]
		upregulating detoxification enzymes	Synthetic	Mouse cortical neurons	[111]
		Chemoprevention	Japanese Daikon	Rats liver and lung	[104]
		Chemoprotection	Synthetic	Human colon cancer cells	[112,113]
		histone deacetylase activity	Broccoli sproutssupplement	Dog	[114]
Glucoraphenin	Sulforaphene	Cytotoxic and cell apoptosis	*R. sativus* seeds	Human hepatocarcinoma HepG2 cells	[115]
Glucoraphasatin	Raphasitin	Enzyme detoxification	*R. sativus*	HepG2 cells	[71]
		Chemoprevention		Human breast MCF-7 cells	[69]
		Chemoprevention	Japanese Daikon	Rats liver and lung	[104]
		Chemo preventive	Daikon sprouts	Rats’ hepatic enzymes	[116]
Gluconapin	3-Butenyl ITCs	Cytotoxicity	*B. juncea* L.	Human prostate cancer cells	[35]
		Cytotoxicity	Synthetic	MCL-5 cells	[99]
		Anti-cancer, radical scavenging ability and increase ROS	*B. juncea* var. *raya*	human breast cancer (MCF-7,MDA-MB-231), prostate cancer (PC-3), lung cancer (A-549), cervical cancer (HeLa) and colon cancer (HCT 116) cells	[100]
Glucoiberin		Cytotoxic activity	*B. oleracea*	colon cancer cells	[56]
Glucocapparin	Methyl ITCs	Chemoprevention	*Capparis spinosa* L.	HT-29 cell	[61]
Glucoarabin, Glucocamelinin		phase II detoxification enzyme induction, quinone reductase	*C. sativa* L. Crantz	Hepa1c1c7 cells	[13]
	α-4-Rhamnopyranosyloxy-benzyl GSs, & isomers	Induction of detoxifying enzymes	*M. oleifera*	human hepatocellular carcinoma cell line -HepG2 cells	[117]
Glucoiberverin, Glucoerucin Glucoiberin		Tumor growth inhibition, Antimicrobial	*L. libyca*	HL60 (human promyelocytic leukaemia cell lines) *C. albicans, P. aeruoginosa*	[36]
**Other activities**					
Sinigrin	Allyl ITCs	Antifibrotic activity	Synthetic		[118]
		Antiglycation	Synthetic		[119]
Glucoraphenin, Glucoraphasatin	Sulforaphene, Raphasatin	Blood sugar control	*R. sativus*		[120]
Glucoraphanin, Glucoerucin	Sulforaphane, Erucin	Suppress mutagenecity	Synthetic		[121]
progoitrin, glucoraphanin, glucoalyssin, gluconapin, glucoerucin, glucoberteroin, glucobrassicanapin		Bone formation	*B. rapa* L.		[66]
**Indole GSs**					
**Biocidal activity**					
Glucobrassicin	Indole-3 carbinol	Biocide	Synthetic		[72]
**Antioxidant activity**					
Glucobrassicin		Antioxidant	*B. rapa* L.		[19]
			Curly kale leaves		[92]
			*Isatis canescens*		[122]
**Chemoprevention**					
Glucobrassicin	Indole-3-carbinol	Chemoprevention	Synthetic	Rats’ hepatic and kidney mice bone marrow cells	[123,124]
	3,3′-Diindolylmethane	Enhances level of reduced-glutathione	Synthetic	Rats heart and bone marrow	[125]
**Other activities**					
Glucobrassicin, neoglucobrassicin		Bone formation	*B. rapa* L.		[66]
**Aromatic GSs**					
Gluconasturtiin	Phenethyl ITCs	Biocide	Synthetic		[72]
		Larvicidal	*Barbarea verna* and *Barbarea vulgaris*	*Mamestra brassicae*	[126]
Glucotropaeolin, Sinalbin	Benzyl ITCs	Biocide	Synthetic		[72]
Glucolimnanthin	3-ethoxybenzyl ITCs	Bio-herbicide Biopesticide	*L. alba*	Lettuce *Meloidogyne hapla, Pythium irregulare* and *Verticillium dahliae*	[49,50]
GMG	Moringin	Insect deterrence	*M. oleifera* seeds	*Pieris napi, Athalia rosae, Pieris brassicae*	[43]
	Phenyl ITCs	Herbicidal	Synthetic	*C. esculentus* weed	[73]
**Antimicrobial activity**					
GMG	Moringin	Antimicrobial	*M. oleifera*	*P. aeruginosa*, *S. aureus*	[44]
Gluconasturtiin	Phenylethyl ITCs	Antifungal	*A. rusticana*	*T. rubrum, T. mentagrophytes, M. canis, E. floccosum*	[80]
		Antimicrobial	*A. rusticana*roots	*C. albicans, F. nucleatum*	[81]
		Antibacterial	Synthetic	*E. coli*, *S. aureus*	[10,127]
		Antimicrobial	Cruciferous plant	methicillin resistant S*. aureus*	[82]
		Antimicrobial	Synthetic *Tropaeoli majoris, A. rusticanae*	Pig aeromonas intestinal bacteria *P. aeruginosa* *H. influenzae, M. catarrhalis, S. marcescens, P. vulgaris* and *Candida* species	[83,84,85,89]
		Antifungal	Synthetic	* S. sclerotiorum *	[77]
Glucotropaeolin	Benzyl ITCs	Bactericidal	Synthetic	*C. jejuni*	[88]
		Antimicrobial	Synthetic	Antimicrobial	[82]
		Antimicrobial		Pig aeromonas bacteria	[89]
		Antifungal	Synthetic	* S. sclerotiorum *	[77]
Glucobarbarin		Larvicidal	*B. verna* and *B. vulgaris*	*M. brassicae*	[126]
*p*-Methoxybenzyl, *p*-hydroxybenzyl GS		Antifungal	*Tropaeolumtuberosum*	*Phytophthora infestans, R. solani*	[128]
5- Phenylpentyl ITCs		Antimicrobial, cytotoxic & antispasmodic	*A. rusticana*	*S. aureus, B. subtilis, B. cereus, S. enterica, P. vulgaris, E. coli, C. albicans Aspergillus brasiliensis*	[59]
**Antioxidant activity**					
GMG		Antioxidant	*M. oleifera*		[64]
**Anti-inflammatory activity**					
	Phenyl ITCs	Anti-inflammatory	Synthetic		[129]
Gluconasturtiin	Phenethyl ITCs	Anti-inflammatory	Synthetic		[130]
GMG	Moringin	Anti-inflammatory	*M. oleifera*		[96,131,132]
3,4-Dimethoxyphenyl		Anti-inflammatory	Synthetic		[27]
	3-methoxyphenyl ITCS	Anti-inflammatory	synthetic		[129]
**Cytotoxicity and chemoprevention**					
GMG		cytotoxicity	*M. oleifera*	human colon adenocarcinoma grade II cells	[133]
		Inhibit cells proliferation		Hep3B Liver Cancer Cells	[41]
		Anticancer		human malignant astrocytoma cell	[40]
Gluconasturtiin	Phenethyl ITCs	Anti-cancer, radical scavenging ability and increase ROS	*B. juncea* var. *raya*	human breast cancer (MCF-7 and MDA-MB-231), prostate cancer (PC-3), lung cancer (A-549), cervical cancer (HeLa) and colon cancer (HCT 116) lines	[100]
		Chemoprevention	Synthetic	Rats’ hepatic and kidney	[123]
		Chemoprevention	Synthetic	prostate cancer cells	[134]
		Anti-cancer, anti-melanoma	*E. sativa* seed oil	HepG2 human liver carcinoma and BB16F10 mice melanoma cell line	[101]
		Inhibit multiple myeloma growth	Synthetic	multiple myeloma cells	[135]
Glucotropaeolin	Benzyl ITCs	Anticancer activity	*Carica papaya* L.	human lung cancer H69 cell	[37]
		Inhibit growth	Synthetic	multiple myeloma cells	[135]
	Benzyl and Phenyl ITCs	Antimetastatic	Synthetic	MDA-MB-231 (breast), Caski (cervical), A549 (lung), and U2OS (osteosarcoma) cell lines	[108]
	A, β-dialkoxyphosphoryl alkyl & aralkyl ITCs	Antiproliferative activity	Synthetic	Lung cancer cells	[30]
	3-Methoxybenzyl ITCs,	Photoprotective	*L. alba*	human skin cells	[48]
	5- Phenylpentyl ITCs	Antimicrobial, cytotoxic & antispasmodic	*A. rusticana*	*S. aureus, B. subtilis, B. cereus, S. enterica, P. vulgaris, E. coli, C. albicans A. brasiliensis*	[59]
**Other activities**					
Gluconasturtiin		Bone formation	*B. rapa* L.		[66]
	Phenethyl ITCs	Suppress mutagenecity	Synthetic		[121]
	Phenyl ITCs	Anti-inflammatory, cholinesterase inhibitory	Synthetic		[129]

**Table 2 molecules-25-03682-t002:** Summary of some preclinical and clinical studies evaluating the health benefits of GSs and GSHPs.

Patient’s Condition	Participants Enrolled	Groups and Doses	Duration	Analysed	Clinical Outcomes	Results	Reference
Melanoma	17	3 groups, 3 oral doses (50,100, 200 µmoles) of broccoli sprouts extract containing sulforaphane (SFN)	28 days	17	sulforaphane levels in plasma and skin, plasma cytokines, Safety, tissue proteomics	Detectable SFN in plasma, cytokines decreased, extracts tolerated up to 200 µmol, increased tumor suppressors	[145]
Type 2 diabetes patients	81	3 groups (2 received 10 g/day (d) broccoli sprout powder (BSP) and 5 g/d BSP and the third received the placebo	4 weeks	72	Insulin concentration, fasting serum glucose, glucose: insulin ratio, homeostasis model of insulin resistance index	10 g/d BSP significantly decreased serum insulin concentration, and improved insulin resistance in patients	[146]
Low and intermediate risk prostate cancer patients	61	3 groups received each 300 mL of different broccoli soup (rich in glucoraphanin) weekly	12 months	48	Tissues examination/prostate pathology and RNA sequencing analysis for gene expression	Reduction of prostate cancer progression. Soups with high glucoraphanin content caused suppression in expression.	[147]
Patients with recurrent prostate cancer	20	200 µmoles/day of sulforaphane rich extract broccoli sprouts	20 weeks	16	Safety of doses, Prostate specific antigen (PSA)% levels	Treatment was safe, exhibited anti-tumor potential, majority of patients didn’t get reduced PSA levels	[148]
Asthmatic patients	51	100 µmoles/day of sulforaphane rich broccoli sprouts extract	14 days	45	Pulmonary functions, NAD(P)H quinone dehydrogenase-1 gene expression, safety	Enhanced bronchoconstrictor effects, increased gene expression also enhanced broncho-protection	[149]
Women with abnormal mammograms	54	2 groups; group1: (2 pills, 3 times/day of 30 mg glucoraphanin), group 2 placebo	2 to 8 weeks	48	Sulforaphane in blood and urine, tissue biomarkers	Safe but not efficient to produce the changes in breast tissue tumor biomarkers	[151]
Schizophrenia	10	3 tablets/30 mg of Sulforaphane per day	8 weeks	7	Evaluation of Symptoms using positive and negative syndrome scale and cognitive function	Sulforaphane potentially improved cognitive function	[95]

**Table 3 molecules-25-03682-t003:** Improving GSs/ GSHPs and bioactivity in plant.

Treatment/factor	Plant	GSs	Biological Activity	Reference
Sodium chloride	Radish sprouts Broccoli sprouts	Total GSs, Glucoraphasatin Sulforaphane	Antioxidant activity -	[161,162]
Salinity and carbon dioxide	*B oleracea*	Indolic GSs, aliphatic GSs respectively	-	[163]
Sulphur salt supplementation	*B. oleracea* Var*. capitata, R. sativus*	Total GSs, progoitrin, glucoerucin, glucobrassicin, glucohirsutin and 4- methoxybrassicin	Antioxidant, anti- proliferative	[68]
MeJA	*B. napus, B.oleracea*	gluconasturtiin, glucobrassicin, neoglucobrassicin, glucoraphanin	Quinone reductase	[164]
	Broccoli florets	glucobrassicin, neoglucobrassicin, gluconasturtiin	Quinone reductase	[169]
	*B. rapa* ssp. *Chinensis* *B. rapa* L. *B. napus* var. *pabularia* *B. oleracea*	1-methoxy-3-indolylmethyl GSs Indole GSs Total GSs, sulforaphane glucobrassicin, neoglucobrassicin, gluconasturtiin	Mutagenecity - - -	[57,166,171,170]
MeJA, high temperature, water stress	Broccoli cultivars	Total GSs	Quinone reductase	[168]
GA3 with glucose	*B. oleracea*	Indolic GSs	Antioxidant	[70]
JA	*B. oleracea* var. *italica*	Glucoraphanin	Not evaluated	[165]
Selenium	Broccoli sprouts *R. sativus*	Sulforaphane Glucoraphanin	Not evaluated	[34] [173]
Nitrogen nutrition (Ammonium)	*A. thaliana* and *B. oleracea*Var. *italica*	GSs	Not evaluated	[174]
Ozone	*Brassica oleracea* var. *capitata* f*. alba*	Sinigrin	Antioxidant	[180]
6-benzylaminopurine	*B. oleracea*	Total GSs and sulforaphane	Antioxidant	[175,176]
Green LED light	*B. oleracea*	Total GSs		[177]
Larval infestation *Delia radicum* root herbivory *D. radicum* and *Delia floralis*	*B. napus* var. *pabularia* *B. rapa* L. *B. rapa* varieties	Indole GSs Benzyl and indole GSs in roots Aliphatic, indole and benzyl GSs in roots of high GSs varieties		[166,178,179]
*BoTGG1* Gene	*A. thaliana*	GSs	Bacterial pathogens attack	[172]

**Table 4 molecules-25-03682-t004:** Improving GSs/ GSHPs bioavailability and bioactivity.

Treatment	Source	Effects	Biological Activity	
Mild heat treatment	*B. rapa* L.	Myrosinase enzyme activity		[182]
Continuous consumption	Broccoli	Total ITCs in colon and caecum levels of male mice	Quinone oxidoreductase	[187]
Exogenous myrosinase treatment	Broccoli sprouts Cooked broccoli Mustard powder	Sulforaphane in diet Sulforaphane in cooked vegetables Enzyme activity	Sulforaphane Plasma levels Urinary sulforaphane levels Antimicrobial	[183,184,186]
Powdered mustard seeds	*B. oleracea* var. *italica*	Sulforaphane in processed vegetables	-	[185]

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
