# Peer review of "Human, Animal and Plant Health Benefits of Glucosinolates and Strategies for Enhanced Bioactivity: A Systematic Review"

_molecules, 2020, doi:10.3390/molecules25163682_

Round 1

Reviewer 1 Report

The paper of Maina et al., is an interesting review on glucosinolates, secondary metabolites worthy to be discussed and better investigated.

The conceivement is original and I find that to consider also the factors that affect GS content and bioactivity is a strenght of the work.

Nevertheless I found some major pitfalls and I have some doubts.

The most important points: 

Results don't show clinical trials and only few animal studies. Why? Actually, many studies are published and should be considered.

Results also don't show results about botanical extracts and herbal preparations standardized in GS, but these ingredients are actually the most used and studied for human health.

Did authors include single phytochemicals names in the research protocol? I find the that the terms (the generic term glucosinolates and other related to the only word glucosinolates) used to construct the database and reported in materials and method could be insufficient and they hardly could be adequate for a comprehensive discussion.

Another important point: 

In 3.8 I truly suggest to enlarge the description: the table is not enough to clearly understand the key points of published papers and many important data are missing.

Not less important:

Some points of the paper are not clear and I suggest to revise spelling and grammar.

For example: 285: this statement is not clear: what do you mean by saying: "GSs chemical composition"?

Minor points:

3: results

16: GSs/GSHPs: the acronym should be explained

37: sulphur

Botanical names:

the first time a name is cited, it should be written in the complete form: Sinapsis alba L. ... etc. then only abbreviated: for example: B. rapa L. change in B. rapa.

Author Response

Thank you very much for your input in our review paper. We appreciate the time, efforts and the points you highlighted to make the document better. Kindly, attached are the file on the responses to the highlighted points.

Thank you in advance from the team.

Sincerely, 

Ho Youn Kim

Reviewer 2 Report

This review is focused on glucosinolates, plant defense-related compounds. Glucosinolates are used by plants for defense against herbivores and pathogens. As bioactive compounds, glucosinolates can be used as drugs, antimicrobial chemicals, pesticides, and food additives. They are produced by plants in the mustard family and natural sources in the human diet include broccoli, cabbages, bok choy, etc. This review gives a good summary of those topics and is a good reference for the variety and plant sources of glucosinolates. The review cites health benefits of glucosinolates in food but there are examples of glucosinolate toxicity in animals and those could be cites for a more balanced coverage of the topic. Including “glucosinolate toxicity” as a search term would be useful.

Minor points:

Line 17: “Only literature” change to “In this review, only literature”

Line 22, 23: “Their improvement utilizes mostly biotic/abiotic stresses and short periods of phyto-hormones application.” Not clear if you mean “Methods for increasing plant glucosinolate content include applying biotic/abiotic stresses and short periods of phyto-hormone application.”

Line 25 “there’s need” change to “there is a need”

Line 150 “Besides, non-volatile ITCs like Moringin facilitates feeding deterrence of Brassicales specialists exhibiting herbicidal activities” The meaning is not clear. Do you mean “Non-volatile ITCs like Moringin facilitate feeding deterrence of Brassicales specialists.” Don’t know what is meant by “exhibiting herbicidal activities”.

Line 161 “Antimicrobial” change to “Microbial”. “Pathogenic resistance” is not a problem, you must mean something else such as “pathogens”.

Line 231 “detoxification of enzymes” is not a clear concept. GS could induce expression, stimulate activity, compete away an inhibitor, etc.

Line 291 NaCl not NaCl2

Author Response

(The authors gave the same response as above.)

Round 2

Reviewer 1 Report

See attachment

Author Response

kindly find attached the response to the points and concerns highlighted in round 2 of the review.
